# Crystal structures of phosphatidyl serine synthase PSS reveal the catalytic mechanism of CDP-DAG alcohol O-phosphatidyl transferases

Martin Centola [1], Katharina van Pee[1], Heidi Betz[1] & Özkan Yildiz [1✉]

Phospholipids are the major components of the membrane in all type of cells and organelles. They also are critical for cell metabolism, signal transduction, the immune system and other critical cell functions. The biosynthesis of phospholipids is a complex multi-step process with high-energy intermediates. Several enzymes in different metabolic pathways are involved in the initial phospholipid synthesis and its subsequent conversion. While the "Kennedy pathway" is the main pathway in mammalian cells, in bacteria and lower eukaryotes the precursor CDP-DAG is used in the de novo pathway by CDP-DAG alcohol O-phosphatidyl transferases to synthetize the basic lipids. Here we present the high-resolution structures of phosphatidyl serine synthase from *Methanocaldococcus jannaschii* crystallized in four different states. Detailed structural and functional analysis of the different structures allowed us to identify the substrate binding site and show how CDP-DAG, serine and two essential metal ions are bound and oriented relative to each other. In close proximity to the substrate binding site, two anions were identified that appear to be highly important for the reaction. The structural findings were confirmed by functional activity assays and suggest a model for the catalytic mechanism of CDP-DAG alcohol O-phosphatidyl transferases, which synthetize the phospholipids essential for the cells.

---

[1] Department of Structural Biology, Max Planck Institute of Biophysics, Max von Laue Strasse 3, 60438 Frankfurt am Main, Germany.
✉email: Oezkan.Yildiz@biophys.mpg.de

Phospholipids are the most important structural components of biological membranes in all living organisms[1]. In eukaryotic cells, the most abundant phospholipids are phosphatidylethanolamine (PE, cephalin) and phosphatidylcholine (PC, lecithin). PE is also the main phospholipid in most prokaryotic cells, whereas PC is absent in most of them[2]. Other important phospholipids in eukaryotic and bacterial cells are phosphatidyl serine (PS), phosphatidic acid (PA), phosphatidylglycerol, phosphatidylinositol (PI), and cardiolipin (CL). The conserved phosphodiester-linked polar head group defines the structural and functional role of the phospholipids in archaea, bacteria and eukaryotes. Although in bacterial and eukaryotic phospholipids the majority of the hydrocarbons are saturated and unsaturated $C_{16}$ or $C_{18}$ fatty acids linked to a glycerol-3-phosphate (G3P) via an ester bond, the isoprenoid hydrocarbon chains in archaeal phospholipids are linked to glycerol-1-phosphate (G1P) via ether bonds[3,4]. In addition to their role in membrane structure, phospholipids also have important functions in cell metabolism, signal transduction, blood clothing, and immune system activation[5–9].

The biosynthesis of phospholipids is a complex multi-step process involving the formation of glycerol-phosphates and other high-energy intermediates such as cytidine-5′-diphosphate (CDP)-choline, CDP-ethanolamine, CDP-diacylglycerol (CDP-DAG) and fatty acids activated by co-enzyme A (fatty acyl-CoA). The reaction of glycerol-phosphate with fatty acyl-CoA forms PA, which can be dephosphorylated to DAG[10]. Two conserved pathways are known so far for the lipid head group transfer. In the so-called "Kennedy pathway," ethanolamine or choline is phosphorylated to phosphoethanolamine or phosphocholine[11], which then react with cytidine-5′-triphosphate (CTP) to form CDP-ethanolamine or CDP-choline and pyrophosphate[12]. In the final step, they condense with DAG to form PE or PC and release cytidine monophosphate (CMP). In mammalian cells, the "Kennedy pathway" is the predominant mechanism for phospholipid synthesis.

In the other pathway that is known as CDP-DAG or de novo pathway, CDP-DAG reacts with an alcohol in a nucleophilic reaction. This reaction is catalyzed by CDP-DAG alcohol O-phosphatidyl transferases (CDP-AP, EC 2.7.8.8) and leads to the synthesis of a lipid molecule and the release of CMP. In some Gram-negative bacteria, the enzymes are peripheral membrane proteins and are associated with the soluble ribosomal fraction[13–15], whereas in Gram-positive bacteria, lower eukaryotes, and plants the enzymes are integral membrane proteins[16,17]. The hydrophobic chain of the substrates, which are geranyl groups (CDP-archaeols) in archaea and acyl groups (CDP-DAG) in bacteria or higher eukaryotes, has no obvious effect on CDP-AP activity[10,18,19]. CDP-APs have a conserved sequence motif DxxDGxxAR(nx)GxxxDxxxD and their activity depends on divalent cations[16,20]. The alcohols most commonly used by the CDP-APs are ethanolamine, serine, glycerol, inositol, and choline. Phospholipids can also be converted by a reversible exchange or modification reactions of their head groups. The enzyme PS-decarboxylase converts PS to PE by decarboxylation[14], whereas the enzyme PE serine transferase exchanges the serine of PS for ethanolamine to form PE[11,21]. Triple methylation of PE by methyltransferase is another important reaction leading to the formation of PC[22].

In contrast to cationic PC and PE, the neutral head group of PS makes the entire lipid anionic and creates an electrostatic charge on the cell surface that attracts cations. In yeast, PS is essential for proper cell polarization and localization of Cdc42[23]. Yeasts lacking PS synthesis are auxotrophic for choline and ethanolamine, and show mitochondrial abnormalities and formation of petite cells[24]. It also leads to a variety of defects in morphology,

cytokinesis, actin cytoskeleton, and cell wall integrity. Overexpression and upregulation of PS synthesis leads to similar defects implying that synthesis of PS must be finely tuned[25]. The constant regulation of asymmetric lipid distribution leads to an accumulation of PS on the inner leaflet of the plasma membrane. In apoptotic cells, this machinery is disrupted and lipid asymmetry is disturbed. The presence of PS on the outer leaflet of the plasma membrane indicates apoptosis and is a signal for macrophages to phagocytose the cell[26,27]. PS has a dual function in the cell as a product or as an intermediate, as it is also a precursor for the synthesis of PE and PC. PS can be decarboxylated to PE by PS-decarboxylase and then further react to PC by triple methylation by phospholipid N-methyltransferase[28].

Structural information on CDP-APs has so far been limited to the di-myo-inositol-1,3′-phosphate-1′-phosphate synthase Af0263 from Archaeoglobus fulgidus (AfDIPPS)[29], Af2299 from A. fulgidus that probably is involved in the synthesis of PI-phosphate[30], PI-phosphate synthases (PIPSs) from Renibacterium salmoninarum (RsPIPS)[31], and PIPS from Mycobacterium tuberculosis (MtPgsA1)[32] and Mycobacterium kansasii (MkPIPS)[33]. In AfDIPPS, a nucleotidyl transferase domain (IPTC) is linked to the N terminus of the DIPPS domain, which consists of six transmembrane helices. In the apo structure of AfDIPPS, these helices form a wide-open substrate-binding site with only one $Mg^{2+}$ ion bound[29]. The apo structure of Af2299[30] is largely identical to the AfDIPPS structure, except that the metal binding site in Af2299 was interpreted as having two divalent metal ions, whereas only one metal ion was found in AfDIPPS. The Af2299 structure was also solved with bound CDP and CDP-glycerol[30]. RsPIPS was solved with bound CDP-DAG and as a chimera consisting of the IPTC domain of Af2299 and the membrane domain of RsPIPS[31]. MtPgsA1 crystallized with CDP-DAG, without substrate and in complex with $Mn^{2+}$ and citrate[32], whereas MkPIPS crystallized in the apo state and with bound IP and CDP. Superimposition of the structures shows that the membrane domain in all of them has the same overall fold with six transmembrane helices that differ slightly in their substrate-binding sites and flexible loops. Nevertheless, the structures provide a first insight into the overall structure of PIP synthases.

The CDP-AP phosphatidyl serine synthase (PSS) plays a crucial role in the virulence of pathogenic fungi. Candida albicans with PSS (CHO1) shows deficiency in PE synthesis, as PS is a precursor to PE. Fungi with mutations in PSS are less virulent and have other defects. In Brucella abortus, mutations in PSS lead to a reduced production of PS, resulting in reduced PC levels and virulence of the pathogen[34]. PS is also important for HIV virus–host interactions, as the HIV envelop is enriched in PS and plays a role in the virulence[35]. Here we present our work on the CDP-AP PSS from M. jannaschii (MjPSS), which catalyzes the formation of PS from CDP-DAG and serine. We purified MjPSS after heterologous expression from Escherichia coli, functionally characterized its enzymatic activity, crystallized it in the lipidic cubic phase (LCP), and solved its structure. Our functional results show that the activity of MjPSS is not restricted to archaeal substrate CDP-archaeol. MjPSS also can perform the CDP-AP reaction with the non-archaeal precursor CDP-DAG and serine to form PS instead of its natural product archaeatidyl-serine. The crystal structure of MjPSS confirms its dimeric state. Unlike PIP synthases with six transmembrane helices, MjPSS has eight helices fully embedded in the membrane. In total, we solved four different structures, with MjPSS in two main conformations: (I) CDP-DAG, $Ca^{2+}$, $Mg^{2+}$, and citrate; (II) CDP-DAG, two serines, $Ca^{2+}$, $Mg^{2+}$; (III) CDP-DAG/serine complex, $Ca^{2+}$, and $Mg^{2+}$; and (IV) CDP-DAG and $Ca^{2+}$. Based on the functional and structural results presented here, we provide a working model for the enzymes that catalyze the CDP-AP reaction, in particular the

important de novo synthesis of PS, which is an essential component of membranes and cellular functions, as well as a precursor for other lipids.

## Results

**Enzyme activity and substrate specificity.** For functional and structural analysis, PS synthase was cloned from *M. jannaschii* (MjPSS) and purified from *E. coli* membranes by immobilized metal affinity chromatography and size-exclusion chromatography. The enzymatic properties of MjPSS were characterized in vitro by high-performance liquid chromatography (HPLC) or high-performance thin-layer chromatography (HPTLC). HPLC was used to quantify the release of CMP after incubation of CDP-DAG and serine in presence of MjPSS. HPTLC was used for the qualitative detection of lipid products and precursors (Supplementary Fig. 1a). MjPSS has an extremely high substrate specificity for serine, whereas other potential substrates found in head groups of typical lipids do not show any reaction (Fig. 1a). They also do not compete for the serine-binding site, as they did not reduce activity when added to the reaction mixture along with serine in competition assays (Supplementary Fig. 2a). Analysis by binding using a microscale thermophoresis assay shows a marked fluorescence change in the presence of CDP-DAG or serine, whereas the fluorescence change of other possible substrates is much lower (Supplementary Fig. 2b). The binding does not appear to be sequential, as both substrates change the fluorescence signal independently of each other.

CDP-APs depend on divalent metal ions for their activity[17,36]. Using atomic absorption spectroscopy, we showed that $Mg^{2+}$ and $Ca^{2+}$ were present in the purified protein solution. The addition of further $Mg^{2+}$ resulted in a slight increase in enzyme activity, whereas the addition of extra $Ca^{2+}$ significantly inhibited the reaction (Fig. 1b). The addition of the divalent cations $Mn^{2+}$ and $Co^{2+}$ showed a similar result such as $Mg^{2+}$, whereas the addition of $Zn^{2+}$ led to a slight increase in activity (Fig. 1b). Removal of the co-purified divalent metal ions by EDTA chelating stops the enzyme activity, whereas EGTA chelating $Ca^{2+}$ with higher affinity has no effect or a light increase in activity (Fig. 1b). The inhibition by EDTA could be completely reversed by the addition of extra divalent metal ions, including $Mg^{2+}$, $Mn^{2+}$, $Co^{2+}$, or $Zn^{2+}$, but not $Ca^{2+}$ (Supplementary Fig. 2c). The addition of $Zn^{2+}$ shows hyperactivation compared to the other divalent cations (Supplementary Fig. 2c). Interestingly, in the presence of $Zn^{2+}$, MjPSS not only catalyzes the synthesis of PS from CDP-DAG and serine, but also the formation of PA and CMP from CDP-DAG, even in the absence of serine (Supplementary Figs. 1b–d and 2c, d). This indicates an irreversible hydrolysis of CDP-DAG by MjPSS in the presence of $Zn^{2+}$. These results show that MjPSS is not specific for divalent cations, as $Mg^{2+}$, $Mn^{2+}$, $Co^{2+}$, and $Zn^{2+}$ can bind and drive the reaction, while $Ca^{2+}$ plays an inhibitory role.

The kinetic parameters of MjPSS activity as a function of its substrate were determined by measuring the release of CMP at different concentrations of serine (Fig. 1c, d). Our findings that both substrates, CDP-DAG and serine, are required for the proper activity of MjPSS suggest a bi–bi reaction mechanism of a second-order nucleophilic attack[16,20,36–38]. If the reaction products PS and CMP are present at higher concentrations, MjPSS can also catalyze the reverse reaction (Supplementary Fig. 1e), similar to PSS of *Saccharomyces cerevisiae*[39]. However, the reverse reaction is not catalyzed when PA is used instead of PS, indicating that the binding of PS by MjPSS is also highly specific. By serine titration, we determined a $K_m$ value of 0.46 mM ± 0.12 and $V_{max}$ of 2.22 μM/s ± 0.13, from which we calculated the turnover rate $k_{cat}$ to be 0.123 s$^{-1}$ ± 0.007. The pH-dependent MjPSS enzyme activity shows a peak at pH ~7 (Fig. 1e) and the temperature optimum in vitro is slightly above 80 °C (Fig. 1f), the environmental temperature of 85 °C of where *M. jannaschii* live[40].

**Overall architecture of MjPSS.** To understand the molecular mechanisms of CDP-DAG alcohol O-phosphatidyl transferase (CDP-AP) activity of MjPSS, crystallization experiments were performed in the presence and absence of substrates or divalent cations. Analysis of all data sets revealed a total of four different structures in two overall conformations (Table 1 and Supplementary Figs. 4 and 5). In the first conformation, native MjPSS crystallized with CDP-DAG, $Ca^{2+}$, $Mg^{2+}$, and citrate (I), or CDP-DAG and $Ca^{2+}$ (II). In the second conformation, Selenomethionine (SeMet) MjPSS crystallized with CDP-DAG, two serines, $Ca^{2+}$, and $Mg^{2+}$ (III), or CDP-DAG/serine complex, $Ca^{2+}$, and $Mg^{2+}$ (IV). In all four structures, the asymmetric unit consists of two MjPSS molecules (Fig. 2, Supplementary Fig. 4), which is consistent with SDS-polyacrylamide gel electrophoresis analysis that indicated MjPSS to be a dimer (Supplementary Fig. 3c). The long axis of the dimer is ~65 Å, whereas its height and width are ~45 Å. Each monomer has a total of eight transmembrane helices and a short horizontal helix (hH) in the N terminus aligned parallel to the long axis of the protein (Fig. 2 and Supplementary Figs. 4 and 6). The height of the molecule and distribution of charged or hydrophobic residues on the outer surface indicate that the dimer is fully embedded in the membrane (Fig. 2 and Supplementary Fig. 5).

All helices, with the exception of helix 8, which is tilted by ~45°, are oriented perpendicular to the membrane. Helices 1–4 span the membrane completely, whereas helix 5 and 6 are about half as long as the others, with the consequence that the N terminus of helix 5 and C terminus of helix 6 are located in the membrane core. A short β-strand between helix 4 and 5, together with a β-strand between helix 6 and 7, forms a short antiparallel β-sheet on the cytoplasmic side, covering the termini of helix 5 and 6 in the membrane core towards the cytoplasm. The dimerization interface of ~4000 Å² is formed exclusively by hydrophobic residues of helix 1, 3, and 4 of each protomer. The short hH in the N terminus also contributes to dimerization through interactions with helix 4 and 6 of the other protomer (Fig. 2 and Supplementary Fig. 6). The regions at the membrane boundaries on the cytosolic and extracellular side, which are most likely in contact with the head groups of the membrane lipids, are positively charged, whereas the interface region in the middle along the dimer axis is strongly negative (Supplementary Fig. 5). Negative charges also predominate on the extracellular side of the protein (Supplementary Fig. 5). The protomers within the four different dimers are symmetrically arranged, with one exception where the substrate in the binding pockets was different (Supplementary Fig. 4). Most of the differences found between the protomers are due to the different conformations of the side chains.

**Substrate binding.** The overall organization of the four structures is quite similar (Supplementary Fig. 4). However, a detailed analysis reveals conformational variability and differences in substrate and ion binding (Figs. 3 and 4, and Supplementary Figs. 5 and 7). The main difference is in the conformation of helix 7. In one conformation, helix 7 interacts with helix 2 and shields the negatively charged groove on the cytoplasmic side from the membrane lipids (Fig. 3 and Supplementary Fig. 5a, c). In the other conformation, the cytoplasmic end of helix 7 is bent by ~15° towards helix 8 and moves by 4 Å, opening a 14 Å gap between helix 2 and 7 that connects the negatively charged interface with the hydrophobic membrane side (Fig. 3

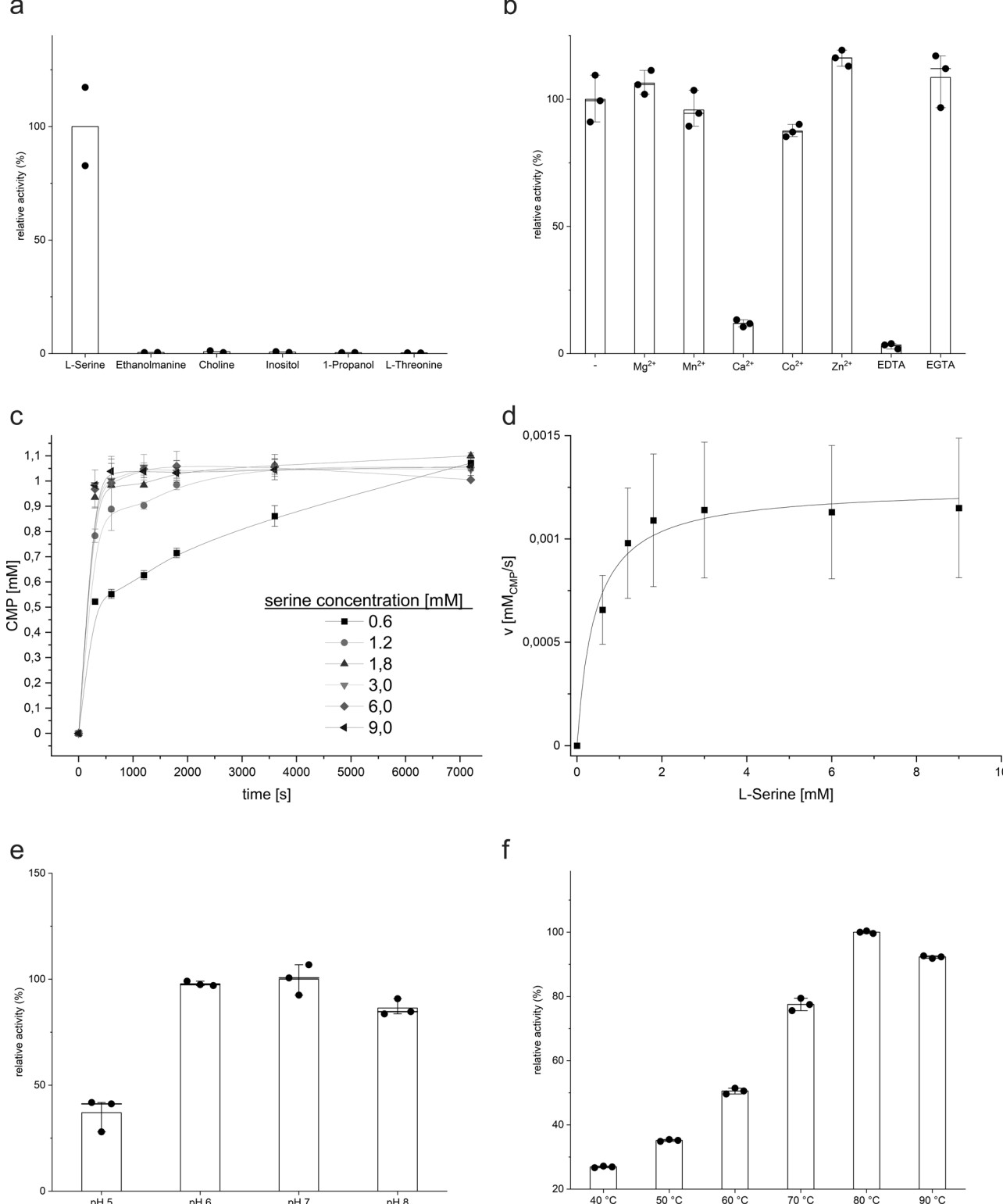

**Fig. 1 Functional analysis of MjPSS. a** MjPSS is highly selective for ʟ-serine, whereas other molecules found in head groups of common lipids show no reaction ($n = 2$, mean). **b** Purified protein has bound $Mg^{2+}$ and $Ca^{2+}$. No change in activity is observed after adding of more $Mg^{2+}$. $Mn^{2+}$ or $Co^{2+}$ also do not change the activity significantly, whereas $Zn^{2+}$ shows a slight increase. Chelation of metal ions with EDTA stops the reaction, while addition of EGTA shows no effect ($n = 3$, mean ± SD). **c** CMP release plotted against time at different initial concentrations of serine ($n = 3$, mean ± SD). **d** Michaelis–Menten plot with initial speed plotted against serine concentration. **e** MjPSS is active in the pH range between 5 and 9, with an optimum at about pH 7 ($n = 3$, mean ± SD). **f** The maximum enzyme activity of MjPSS is at ~80 °C ($n = 3$, mean ± s.d.).

**Table 1 Data collection and refinement statistics.**

| | Closed state (citrate bound) pdb-id: 7B1K | Open state (two serines bound) pdb-id: 7B1L | Transition state (open) (one serine bound) pdb-id: 7POW | Closed state (serine free) pdb-id: 7B1N |
|---|---|---|---|---|
| **Data collection** | | | | |
| Space group | P22$_1$2$_1$ | P22$_1$2$_1$ | C2 | P22$_1$2$_1$ |
| Unit cell (Å) (°) | 62.3, 70.7, 95.4, 90, 90, 90 | 61.63, 79.48, 95.42, 90, 90, 90 | 107.35, 61.22, 79.31 90, 118.72, 90 | 62.13, 70.81, 94.09, 90, 90, 90 |
| No. molecules/AU | 2 | 2 | 2 | 2 |
| Resolution range (Å) | 46–2.2 (2.3–2.2) | 50–1.852 (1.92–1.85) | 38–2.5 (2.6–2.5) | 50–2.8 (2.9–2.8) |
| Wavelength (Å) | 0.99989 | 0.9999 | 0.9796 | 1.00 |
| X-ray source | SLS X10SA | SLS X10SA | Petra3 P11 | SLS X10SA |
| $R_{merge}$ | 0.20 (2.71) | 0.106 (1.42) | 0.15 (0.29) | 0.243 (1.92) |
| $R_{meas}$ | 0.22 (2.93) | 0.116 (1.57) | 0.16 (0.31) | 0.266 (2.11) |
| $R_{pim}$ | 0.08 (1.07) | 0.05 (0.656) | 0.05 (0.12) | 0.107 (0.864) |
| CC1/2 | 0.998 (0.389) | 0.999 (0.792) | 0.987 (0.968) | 0.989 (0.578) |
| CC* | 1 (0.748) | 1 (0.94) | 0.997 (0.992) | 0.997 (0.856) |
| Mean $I/\sigma(I)$ | 6.9 (0.8) | 9.36 (1.35) | 12.9 (6.02) | 4.51 (0.86) |
| Wilson B-factor (Å$^2$) | 46.83 | 23.89 | 24.92 | 61.44 |
| Completeness (%) | 97.67 (96.86) | 97.51 (95. 63) | 99.84 (99.55) | 99.12 (97.60) |
| Total reflections | 155,009 (14939) | 242,588 (20,434) | 162,723 (10,286) | 66,349 (5839) |
| Unique reflections | 21568 (2098) | 39735 (3636) | 15598 (1547) | 10650 (1017) |
| Multiplicity | 7.2 (7.1) | 6.1 (5.3) | 10.4 (6.6) | 6.2 (5.7) |
| **Refinement** | | | | |
| Reflections (work) | 21514 (2096) | 39610 (3828) | 15591 (1547) | 10607 (1016) |
| Reflections (free) | 1078 (105) | 1978 (190) | 779 (77) | 530 (51) |
| $R_{work}$ | 0.265 (0.375) | 0.2633 (0.393) | 0.215 (0.19) | 0.303 (0.419) |
| $R_{free}$ | 0.294 (0.421) | 0.301 (0.389) | 0.276 (0.22) | 0.358 (0.428) |
| CC$_{work}$ | 0.93 (0.52) | 0.925 (0.871) | 0.894 (0.941) | 0.93 (0.67) |
| CC$_{free}$ | 0.92 (0.59) | 0.807 (0.813) | 0.844 (0.878) | 0.86 (0.49) |
| RMS (bonds) | 0.003 Å | 0.008 Å | 0.008 Å | 0.004 Å |
| RMS (angles) | 0.77° | 1.00° | 0.93° | 0.75° |
| Ramach. favoured (%) | 95.4 | 96.21 | 97.76 | 93.94 |
| Ramach. allowed (%) | 4.4 | 3.54 | 2.24 | 5.3 |
| Ramach. outliers (%) | 0.2 | 0.25 | 0.0 | 0.76 |
| Rotamer outliers (%) | 0.0 | 0.29 | 1.16 | 1.78 |
| No. atoms in AU | 3534 | 3552 | 3613 | 3228 |
| Macromolecules | 3083 | 3083 | 3140 | 3086 |
| Ligands | 285 | 356 | 399 | 130 |
| Solvent | 35 | 113 | 74 | 12 |
| Average B-factor | 61.8 | 30.84 | 31.94 | 69.8 |
| Macromolecules | 59.6 | 28.38 | 31.27 | 69.3 |
| Ligands | 87.6 | 49.36 | 45.42 | 80.4 |
| Solvent | 61.2 | 44.33 | 35.57 | 64.7 |
| Protein residues | 400 | 400 | 406 | 400 |

Statistics for the highest-resolution shell are shown in parentheses.

and Supplementary Fig. 5e, g). Therefore, in the following, we refer to the conformation of the two structures in which helix 7 shields the substrate-binding site from the membrane lipids as "closed" and the second conformations as "open" (Supplementary Figs. 4 and 5). In all structures, the extended electron density within the molecules could be modelled as CDP-DAG, one of the two substrates required by MjPSS for PS synthesis. CDP-DAG was present in all crystallization conditions, as MjPSS did not crystallize without CDP-DAG. The CDP-DAG molecule extends horizontally across the cytoplasmic face of the entire protomer (Fig. 3). In the closed conformation, the two alkyl chains of the DAG moiety are embedded in a hydrophobic channel formed by the side chains of helix 5 and 8 on the one side and helix 7 on the other (Fig. 3 and Supplementary Fig. 5d). The hydrophobic channel has a diameter of ~7 Å and extends from the membrane core at the extreme end of the dimer to the negatively charged groove on the cytoplasmic side (Fig. 3 and Supplementary Fig. 5d). This groove is formed by the interfacial region of both protomers and connects their substrate-binding sites, creating a connection or pathway for potential soluble substrates to enter or access the active site (Supplementary Fig. 5a, e). The DAG portion in

the open structures is branched at the glycerol. One alkyl chain adopts almost the same conformation as in the closed conformation and is enclosed in the hydrophobic channel (Fig. 3, Supplementary Fig. 5c). The other alkyl chain reaches the inside of the membrane through the groove on the molecule side between helix 2 and 7, formed by a 5 Å tilt of the N terminus of helix 7 (Fig. 3 and Supplementary Fig. 5e, g, h). In this conformation, the hydrophobic channel is narrowed to a diameter of ~5 Å at the outer end (Fig. 3 and Supplementary Fig. 5h). The hydrophobic furrow on the molecule side is bounded by positive residues on the cytoplasmic side (Supplementary Fig. 5). Additional hydrophobic and ionic interactions of residues located in the loop before β1, in the loop after β2, and at the N terminus of the short helix 5 contribute to the coordination of CDP-DAG. In all conformations, the CDP moiety is coordinated by highly conserved residues (Supplementary Fig. 8) from helix 1, 2, and 3 and the cytoplasmic loop connecting helix 2 to 3 (Figs. 2 and 3).

Serine, the second substrate of MjPSS, was also needed to obtain larger crystals of high quality. However, as MjPSS did not crystallize when serine was added with additional Ca$^{2+}$, we

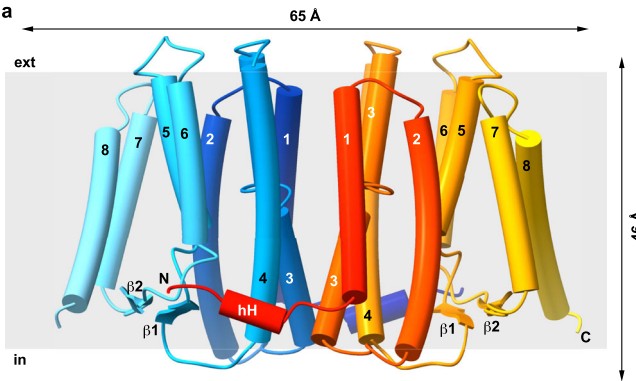

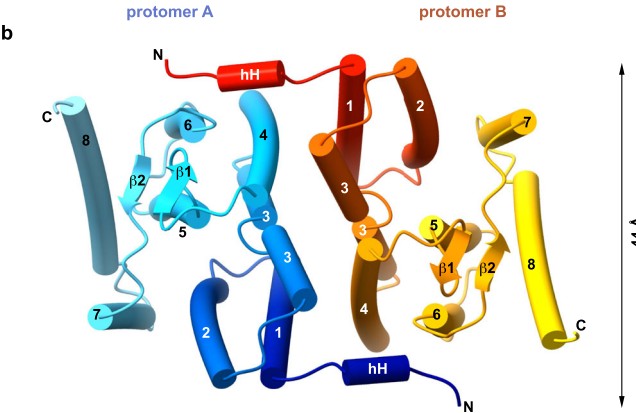

**Fig. 2 Overall structure of MjPSS.** Side view (**a**) and cytoplasmic view (**b**) of the MjPSS dimer. The dimensions of the molecule are indicated by the arrows above and on the right side of the molecule. Protomer A is coloured from blue (N terminus) to cyan (C terminus) and protomer B from red to yellow. The two termini with the short horizontal helix (hH) and the short β-sheet formed by β1 and β2 are located on the cytoplasmic side of MjPSS. The transmembrane helices are numbered from 1 to 8.

omitted serine from the crystallization experiments with $Ca^{2+}$. We were able to localize serine only in the open structures obtained with SeMet-labelled protein, but not in the closed structure obtained from native protein with added serine. Instead, in this structure we found a citrate molecule near CDP-DAG (Figs. 3, 4a). In the structure obtained from crystals with added $Ca^{2+}$ in the absence of serine, no citrate was present (Fig. 4b), although the citrate concentration was the same under all crystallization conditions. As the residues coordinating the citrate molecule (Asn109, Lys111, and Lys157) are not conserved in PSS from diverse organisms (Supplementary Fig. 8), we assume that citrate binding is not specific and consider it as crystallization artefact. In one open structure, we found one tightly bound serine ($Ser_1$) coordinated by residues in helix 3 and 4 and by the ribose unit of CDP-DAG, which points with its hydroxyl group to the β-phosphate of CDP-DAG (Figs. 4c and 5a). All residues (Asp62, Ser63, Asp66, and Phe105) coordinating $Ser_1$ are highly conserved (Supplementary Fig. 8). A second serine ($Ser_2$) was located on the opposite side of CDP-DAG in the hydrophobic groove above the alkyl chain of CDP-DAG, pointing with its hydroxyl group to the glycerol moiety (Fig. 4c). In contrast to $Ser_1$, none of the $Ser_2$-coordinating residues, with the exception of Arg49, is conserved (Supplementary Fig. 8). However, the conservation of Arg49 is apparently due to its role in the coordination of CDP-DAG and charge compensation of the negatively charged CDP-DAG phosphates. The lower position of $Ser_1$ in the molecule therefore suggests that $Ser_1$ is rather the catalytically important serine for the reaction with CDP-DAG. In

the second open structure, we found only one serine in the binding site for $Ser_1$. A closer examination of the electron density of this serine shows a continuous density between the hydroxyl oxygen of $Ser_1$ and CDP-DAG β-phosphate (Figs. 4d and 5b). In that structure, a tetragonal β-phosphate does no longer match the corresponding electron density. Instead, the density indicates a trigonal bipyramidal shape of the β-phosphate (Figs. 4d and 5b), reminiscent of the transition state of the nucleophilic reaction. On the other hand, serine and CDP-DAG are not linked in the second protomer of this structure, as is the case with the monomers of the other open structure.

**Ion coordination**. We interpret most of the spherical electron density blobs in close contact to $Ser_1$ and CDP-DAG phosphates as water molecules. However, the coordination geometry of two of them does not fit well with water molecules and apparently represent two metal ions. The coordination geometry of pentagonal bipyramid for one ion near CDP-DAG matches well with $Ca^{2+}$ ions[41] (Fig. 4). The other metal ion has a slightly distorted octahedral coordination geometry and is best explained by an $Mg^{2+}$ ion (Fig. 4a, c, d). In the closed structure obtained with additional $Ca^{2+}$, the electron density at the coordination site of the $Mg^{2+}$ ion was not clearly interpretable, so we modelled a water molecule into the density at this position (Fig. 4b). In all structures, the $Ca^{2+}$ ion is closely coordinated with the α- and β-phosphates of CDP-DAG, the carboxyl side chains of Asp41, Asp44, Asp62, and the oxygen in the mainchain of Asp41. All three $Ca^{2+}$-coordinating aspartates are part of the highly conserved (Supplementary Fig. 8) sequence motif ($^{41}$DxxDGxxAR(x$_8$)GxxxDxxxD$^{66}$) that defines CDP-alcohol phosphatidyl transferases[42]. The $Mg^{2+}$ ion also is coordinated by side and main chains of Asp62 and the side chains of Asp41, Ser65, and Asp66 from this motif. The presence of the $Ca^{2+}$ and $Mg^{2+}$ ions in the protein solution and crystals were also confirmed by atom absorption spectroscopy (AAS) of purified protein. By AAS, by X-ray fluorescence spectroscopy of protein crystals using synchrotron radiation and by anomalous processing of X-ray diffraction data, we rule out the presence of other cations such as $Zn^{2+}$, $Fe^{2+}$, $Co^{2+}$, or $Mn^{2+}$, which could be bound to and co-purified with MjPSS.

Our structures reveal the importance of the four aspartates in the CDP-AP sequence motif of MjPSS, which mainly coordinate the $Mg^{2+}$ and $Ca^{2+}$ ions directly or via water molecules (Fig. 4). We tested their significance by mutating them to serine or asparagine (Supplementary Fig. 9a). Any mutation of the conserved residues in the CDP-AP motif leads to a significant reduction or complete loss of the enzymatic activity of MjPSS. Some mutations such as D41S, D62S, and D66N alter the cation dependence of MjPSS as the addition of further $Mg^{2+}$ to the reaction mixture partially restores the loss of activity caused by the mutation. These residues, together with Ser65, coordinate directly the $Mg^{2+}$ ion. In comparison, reactivation is negligible or the loss of activity cannot be restored when residues that coordinate both the $Ca^{2+}$ and the $Mg^{2+}$ ion are mutated (D44S, D62N, and D66S; Supplementary Fig. 9a). The structure of MjPSS obtained with $Ca^{2+}$ addition largely corresponds to the closed conformation (Fig. 4a, b). The only significant difference between the two closed structures is the conformation of the loop between helix 4 and β1, and the absence of the citrate molecule (Supplementary Fig. 7c). Upon addition of calcium, the Val110-side chain in this loop occupies the citrate position found in the other closed structure. Activity measurements as a function of citrate show that binding of citrate reversibly inhibits MjPSS, and that the inhibition is reversed by addition of more serine (Supplementary Fig. 10), indicating that presence of citrate

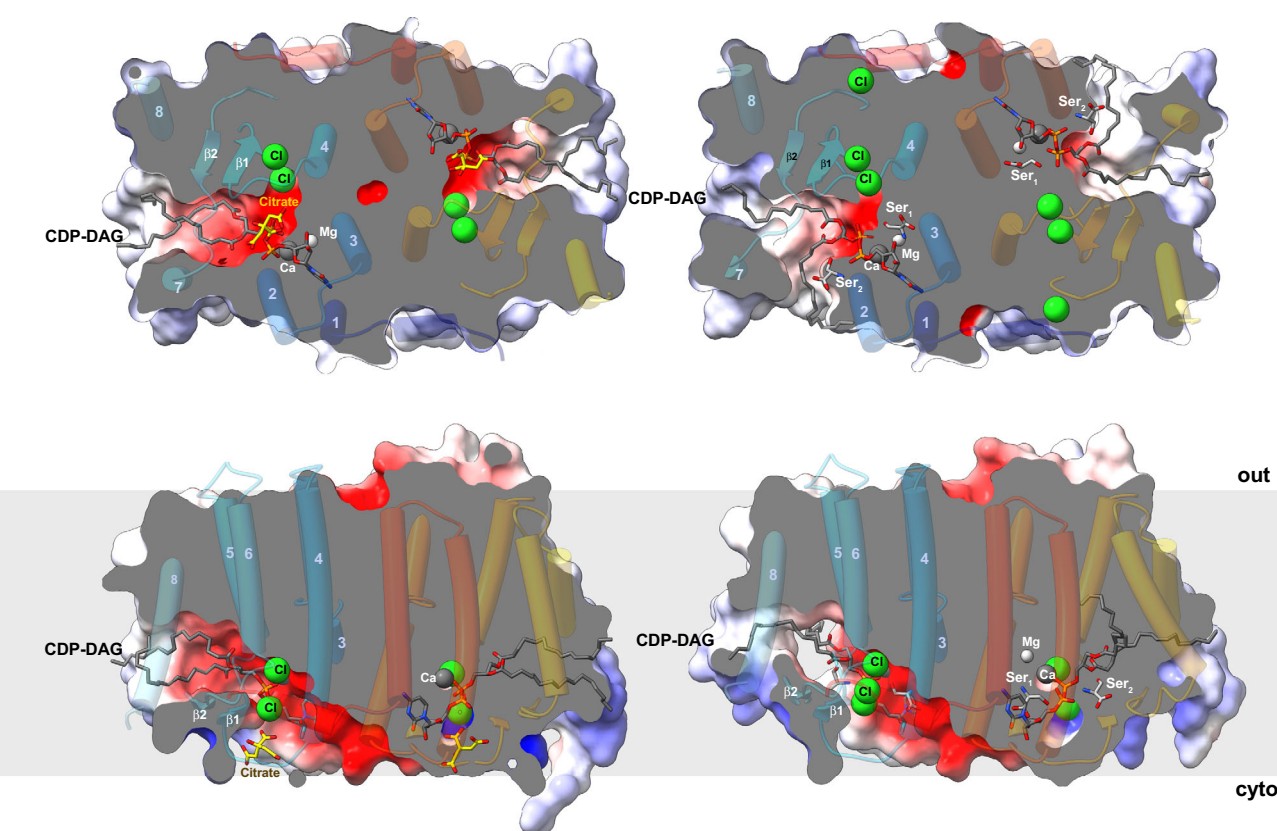

**Fig. 3 Substrate binding by MjPSS.** The large binding pocket for CDP-DAG in MjPSS extends from the hydrophobic membrane core to the active site near the cytoplasmic surface in the centre of the dimer. In the closed structures (left), both CDP-DAG alkyl chains adopt similar conformations within the binding pocket, whereas the positions of helix 7 and 8 in the open structures (right) allow one alkyl chain to reach the membrane via a different path. Serine molecules are only found in the open structures (right). In one closed conformation, there is a citrate near the substrate-binding site, whereas in the other closed structure this position is empty. A chain of three chloride ions extends parallel to the dimer interface from the active site to the cytoplasmic interface with the N-terminal helix hH from the other protomer.

probably blocks the access of serine to the active site or competes with serine for it. Mutation of the residues coordinating the citrate molecule (Asn109, Lys111, and Lys157) to alanine showed no significant change in activity or substrate specificity (Supplementary Fig. 9b).

Further analysis of the immediate vicinity of the active site reveals the presence of two to three chloride ions per protomer (Fig. 3 and Supplementary Figs. 4 and 11). The $Cl^-$ ions, together with the conserved arginines Arg101 and Arg104, form an ionic network in the hydrophobic environment extending from the active site at the position of $Ser_1$ and $Mg^{2+}$ ion to the cytoplasmic interface region of helix 4–6, β1, and the short helix hH from the other protomer (Fig. 3). The high sequence conservation of the residues surrounding the $Cl^-$ anions and the arginine side chains (Supplementary Fig. 8) suggests an important role of the $Cl^-$ ions in MjPSS activity, especially due to their proximity to the catalytically active serine and $Mg^{2+}$. Activity measurements after mutating Arg101 and Arg104 to an aspartate show their importance for the enzyme. The R101E mutation completely abolishes the activity of MjPSS and even the addition of $Mg^{2+}$ cannot restore the activity. The R104E mutation reduces activity by about 50% but the addition of $Mg^{2+}$ compensates for the loss of activity. Similar to the R101E single mutation, the R101E/R104E double mutation leads to a complete loss of activity that cannot be restored by additional $Mg^{2+}$ (Supplementary Fig. 9a). In this context, the short hH at the N terminus seems not only to play a role in dimerization, but is probably also involved in the coordination of the third chloride ion of the ionic network

mentioned above. To test its role, we truncated the first seven residues in the N terminus of MjPSS (MjPSS$_{\Delta N}$) for functional characterization. Although this mutant was unstable and prone to aggregation, we were able to purify and crystallize it. However, the diffraction of these crystals was not good enough for data collection and further optimization of the crystals was not possible. Although MjPSS$_{\Delta N}$ was less stable than the wild-type protein, we were able to measure its activity, which is reduced to about 8% of wild-type MjPSS (Supplementary Fig. 9), indicating an important role of the N-terminal helix.

**Structural comparison of MjPSS and other CDP-APs.** Structural comparisons show that helices 1–6 of MjPSS correspond well to the six transmembrane helices of Af2299, MtPgsA1, and RsPIPS (Supplementary Fig. 11a). However, membrane helices 7 and 8 of MjPSS have no counterparts in the other structures. In MjPSS, helices 7 and 8, together with the short β-sheet, completely shield the substrates from the membrane and partly from the cytoplasm to create a flexible hydrophobic cavity inside the molecule for the alkyl chains of CDP-DAG. In all structures of Af2299, AfDIPPS[29,30], of RsPIPS[31], and MtPgsA1[32], which differ little in the membrane domain with the six transmembrane helices, the substrate-binding pocket is exposed to the membrane and cytoplasm and is only partially covered by the cytoplasmic IPTC domain, if present. They also all have the same dimerization scheme, with the interface between the two protomers being formed across helix 3 and 4 (Supplementary Fig. 11b). In

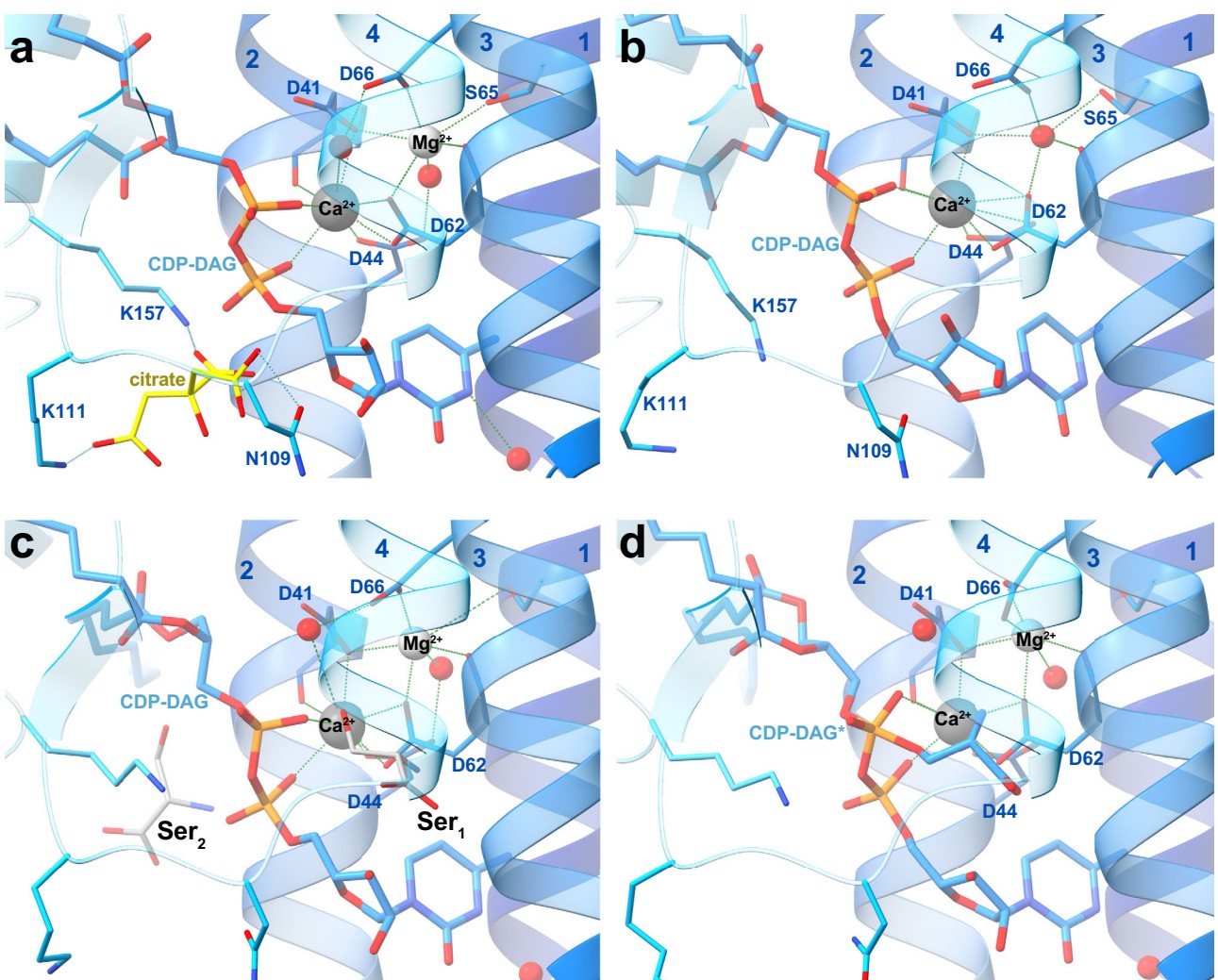

**Fig. 4 Substrate coordination in the MjPSS binding pocket. a** MjPSS in the closed conformation of the native protein with bound CDP-DAG, $Ca^{2+}$, $Mg^{2+}$, and a citrate molecule in the substrate-binding pocket. The citrate molecule is coordinated by Asn109, Lys111, and Lys157, and coordination of the metal ions and CDP-DAG is mediated by the highly conserved residues Asp41, Asp44, Asp62, and Asp66. **b** When MjPSS is crystallized with additional $Ca^{2+}$, it adopts the same closed conformation as in the other closed native structure (**a**). In this structure, $Ca^{2+}$ is in the same position, but the density at the position of $Mg^{2+}$ and the coordination geometry could not prove the presence of $Mg^{2+}$ with certainty, so we modelled this density as a water molecule. **c** The metal coordination of MjPSS in the open conformation resembles the native structure. In this structure, two serine-binding sites are present. In the $Ser_1$ site formed by helix 2, helix 3, CDP-DAG, $Ca^{2+}$ and $Mg^{2+}$ ions, Ala59 and Ser63, the serine is in the optimal position to react with the β-phosphate of CDP-DAG, whereas the position of the serine in the $Ser_2$ site is less optimal for the reaction. The $Ser_2$ site is located near the site where the citrate is non-specifically bound to MjPSS (**a**). **d** In the second open structure, the serine in the $Ser_1$ site is covalently linked to the CDP-DAG in one protomer that probably represents the transition state, whereas in the second protomer serine and CDP-DAG are not linked.

contrast, the dimerization interface in MjPSS is much larger involving helices 1, 3, 4, and the N-terminal short hH (Fig. 2 and Supplementary Fig. 11). The difference in the dimerization interface also leads to a difference in the relative orientation of the protomers to each other (Supplementary Fig. 11b). A hH corresponding to the hH in MjPSS is also present in the structures described above, but it points to the opposite direction and does not contribute to the dimerization of the protomers (Fig. 2 and Supplementary Fig. 11).

We also compared the substrate-binding pockets of MjPSS with bound substrates and ions to the structures of the other CDP-APs (Supplementary Fig. 12). Among the different substrate-binding states, the structures of PgsA1 (pdb-id 6h59) and RsPIPS (pdb-id 5d92) bound to CDP-DAG and the structure of Af2299 bound to CDP-glycerol (pdb-id 4q7c) share a similar arrangement of conserved residues around the CDP moiety (Supplementary Fig. 12). However, with regard to metal

coordination, there are significant differences in interpretation. At the corresponding $Ca^{2+}$ and $Mg^{2+}$ site that we found in MjPSS, two $Ca^{2+}$ ions were modelled in Af2299 (Supplementary Fig. 12a). One is coordinated by the α- and β-phosphates of CDP-glycerol and by the aspartates of the CDP-AP motif, whereas the second $Ca^{2+}$ is coordinated only by the aspartates of the CDP-AP motif-like $Mg^{2+}$ in MjPSS. In MtPgsA1, the equivalent positions were interpreted both as $Mg^{2+}$ ions, but the coordination is similar despite the different metal ions (Supplementary Fig. 12b). Two $Mg^{2+}$ ions are also bound in RsPIPS, but here the α- and β-phosphates of CDP-DAG coordinate the two $Mg^{2+}$ ions directly together with the aspartates from the CDP-AP motif (Supplementary Fig. 12c). The structures also have in common that negatively charged molecules ($SO_4^{2-}$, tartrate) are located near the substrate-binding pocket. These binding sites correspond approximately to the position of the $Cl^-$ ions in MjPSS. However, unlike MjPSS, none of the other structures have both the acceptor

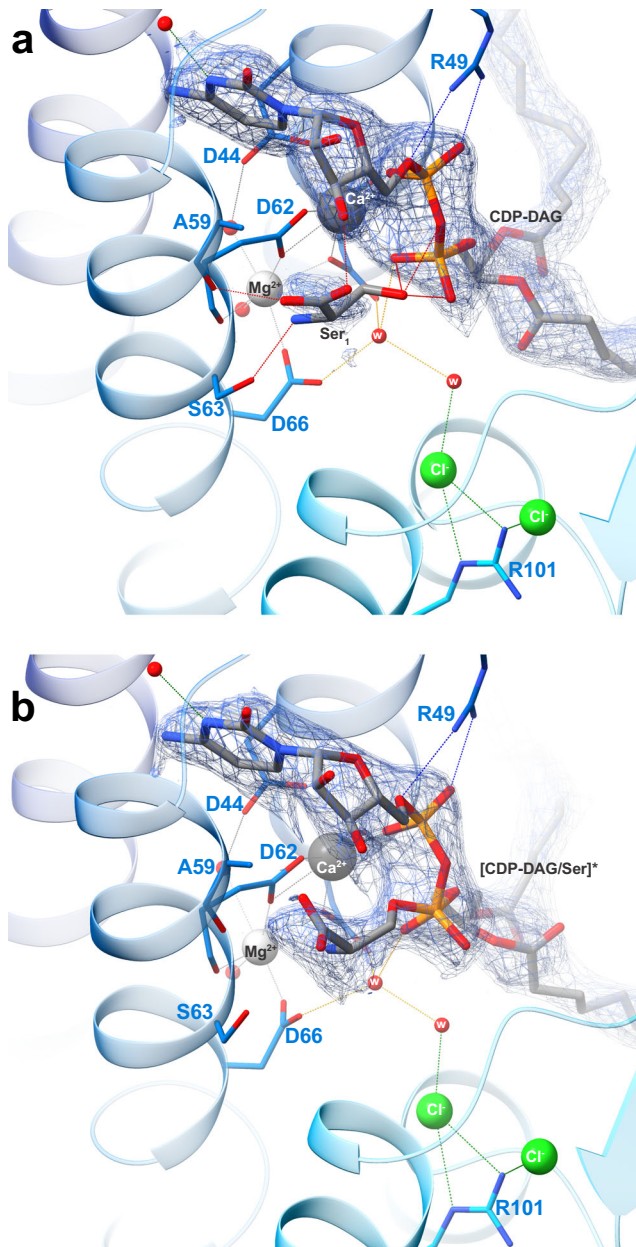

**Fig. 5 Substrate-binding pocket with bound substrates.** Electron density map (2Fo-Fc at ~1$\sigma$) around CDP-DAG and serine in the substrate-binding pocket. In one open structure (**a**), there is no electron density between serine and β-phosphate, whereas in the other (**b**), the serine density is clearly connected to that of β-phosphate, which appears to be in the transition state, as indicated by the flat shape of the surrounding density. In this conformation, the interaction of the β-phosphate with $Ca^{2+}$ has been lost and only α-phosphate coordinates the $Ca^{2+}$ ion. In all structures, $Cl^-$ ions coordinated by the conserved Arg101 interact indirectly via water molecules with serine and CDP-DAG in the substrate-binding pocket.

and donor substrates bound simultaneously, so the presence of negatively charged molecules led to the speculation that they would be the binding site for the donor molecule. However, with the exception of the structure of MtPgsA1 with bound CDP-DAG, where an $SO_4^{2-}$ ion is only 3 Å away from the serine in MjPSS, the positions of the other anions do not match with the position of serine—the donor molecule of MjPSS. As in MjPSS, the residues coordinating the anions are also conserved in these homologous enzymes (Supplementary Fig. 13), indicating a

prominent role for these anions. Interestingly, the anions are also coordinated by conserved residues from the second protomer (Supplementary Fig. 13) and have a potential role in dimerization. In MjPSS, the dimerization creates a cytoplasm-exposed trench with highly polar group (Fig. 3 and Supplementary Fig. 5a, e). This highly charged and solvent-exposed cleft is most probably the entry and biding site for serine.

## Discussion

PS is an important molecule for all kinds of cells. Not only does it serve as a structural component for cellular lipid membranes or as precursor for other phospholipids such as PE, or PC, but it is also essential for signal transduction, cellular metabolism and is essential for viability of many human pathogens[36,43]. Enzymes that synthesize, convert or regulate PS are functionally conserved between bacteria, archaea, and eukaryotes. In the present study, we show that PS synthase from the methanogenic archaeon *M. jannaschii* (MjPSS) is functionally able to catalyze the CDP-AP reaction of serine with CDP-DAG instead of CDP-2,3-diger-anylgeranyl sn-G-1-P, which is the natural substrate of MjPSS[19]. This shows that MjPSS is not specific for the hydrophobic alkyl chain of the phospholipid precursor. The functional findings we present here are also well supported by the structures we solved in near-native lipid environments. The residues that form the hydrophobic cavity for the alkyl chains of the substrate do not exhibit sequence conservation, but are similar in their hydro-phobic nature (Supplementary Fig. 8) and therefore should not distinguish the phospholipid precursors on the basis of their hydrophobic tale. Also the residues involved in the coordination of the glycerol moiety are not conserved, which explains why MjPSS does not discriminate between G1P or G3P lipids. In contrast, the residues coordinating the hydrophilic part of CDP-DAG and serine are highly conserved (Supplementary Fig. 8). This conservation is reflected by the extreme substrate selectivity of MjPSS for serine. A similar selectivity on the head group of the lipid was observed for the archaeon *Methanospirillum hungatei* CL synthase, which was able to catalyze both, the synthesis of CL and glycerol-di-archaetidyl-CL[44]. Moreover, it was able to syn-thetize similar amounts of hybrid glycerol-archaetidyl-phospha-tidyl-CL, and that the reaction was reversible, as we also observed for MjPSS (Supplementary Fig. 1e).

We assume that the cytoplasm-exposed cleft between the two protomers is the entry site for serine into the active site of MjPSS (Fig. 3 and Supplementary Fig. 5a, e). No other potential substrate we tested could functionally replace the serine. Again, the structures with bound serine provide a clear explanation for the high specificity of MjPSS for serine, as we were able to clearly assign the binding site for the catalytically active serine (Ser1). Even threonine, which is structurally most similar to serine, would not fit into the binding pocket without significant rear-rangement of the surrounding residues. The orientation of serine at the Ser1 position is optimal for the nucleophilic attack to the β-phosphorus of CDP-DAG. We consider the second potential serine-binding site (Ser2) to be an experimental artefact due to the relatively high serine concentration during crystallization. The presence of an electron density between Ser1 and the β-phosphate of CDP-DAG in one of the four structures we solved is another strong indication of the correct localization of the catalytically active serine.

Although several crystal structures of CDP-APs have been solved in recent years, the catalytic mechanism of the reaction has not yet been elucidated in detail, because the resolution of the substrate-bound structures were too low or none of the structures were bound to all of their substrates. So far, the reaction mechanism for CDP-APs has been best described for PIP

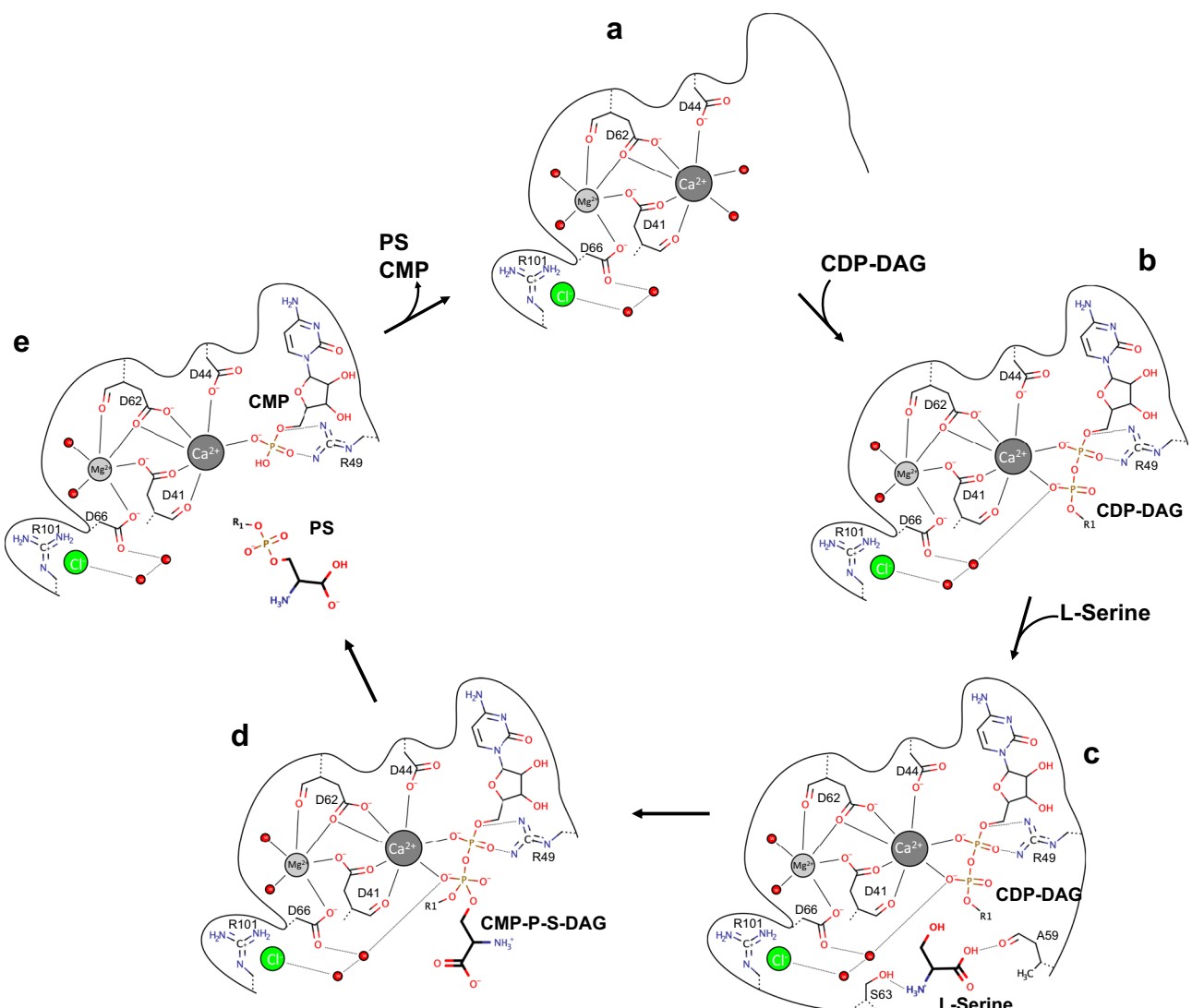

**Fig. 6 Reaction mechanism.** Reaction cycle of MjPSS during the synthesis of PS from CDP-DAG and serine in the presence of $Mg^{2+}$ and $Ca^{2+}$. The CDP-DAG binding site in MjPSS of is formed and stabilized by the divalent cations $Mg^{2+}$ and $Ca^{2+}$ (**a**). In the absence of CDP-DAG, $Ca^{2+}$ most likely is coordinated by water molecules, as shown in **a**, or by residues in nearby loops that would be flexible in the absence of CDP-DAG. The binding of CDP-DAG (**b**) is driven by the coordination of $Ca^{2+}$ by the negatively charged phosphates. Serine binds to the binding pocket after CDP-DAG (**c**). For the nucleophilic attack, the serine molecule is positioned with its hydroxyl group near the β-phosphate of CDP-DAG. The serine molecule is activated by deprotonation, attacks the β-phosphate and forms the penta-coordinated transition state (**d**). The proton of serine is probably removed by one of the water molecules located in the interaction network of Asp66, Arg101, and the nearby chloride ions. Hydrolysis of the CDP-DAG/serine complex from the transition state leads to the complex of MjPSS with the products PS and CMP (**e**). The next cycle starts after release of the products and binding of CDP-DAG. Structural data are available for the state with bound CDP-DAG (**b**), CDP-DAG, and serine (**c**), and for the transition state of the CDP-DAG/serine complex (**d**).

synthesis by MtPgsA1 as it was crystallized with CDP-DAG, which also was resolved in the electron density and in the absence of its substrates[32]. With the new insights we have gained from our structural and functional data, we are able to describe the key steps of the reaction that CDP-APs catalyze in more detail, although structural data for other states, e.g., the apo state of MjPSS (Fig. 6a), are still missing. The serine-bound structure provides important information how the substrates CDP-DAG, serine, $Ca^{2+}$, and $Mg^{2+}$ are bound and oriented relative to each other (Figs. 4c, 5a, and 6b, c). The hydroxyl side chain of serine points directly to the phosphorous atom of the β-phosphate. The lack of electron density between serine and CDP-DAG indicates that serine is not deprotonated. For the next step, the nucleophilic attack on the β-phosphorus of the CDP-DAG, serine must be more nucleophilic than in its free form. One would expect the hydroxyl side chain of the serine to be deprotonated by a nearby

basic residue. One potential candidate, the last aspartate (Asp66 in MjPSS and Asp93 in PgsA1) in the CDP-AP sequence motif[42], has already been proposed[32]. In MjPSS, Asp66 coordinates the $Mg^{2+}$ ion and, via a water molecule, the β-phosphate of CDP-DAG (Figs. 5 and 6), and it is likely that Asp66 increases the basicity of this water in such that it becomes the actual proton acceptor for the serine hydroxyl proton. In addition, the water molecule is also coordinated by Asp41 and is in close contact with the nearby chloride ion, which would further increase the pKa. In the next step, the serine attacks the β-phosphorus with its nucleophilic side chain and transforms the β-phosphorus from the tetragonal to the penta-coordinated β-phosphorus with trigonal bipyramidal coordination geometry, which most likely corresponds to the transition state represented by our structure (Figs. 4d, 5b, and 6d). In the transition state, the β-phosphate also loses coordination with the $Ca^{2+}$ ion. This would support our

assumption above that the proton is transferred from the serine to a water, so that the coordination with $Ca^{2+}$ is compensated by this protonated water, and that the transition of the β-phosphate from the tetragonal to the trigonal bipyramidal conformation is stimulated. The presence of anionic molecules, such as $Cl^-$ in MjPSS, around the active site would lower the pKa of serine and make it easier for the serine to release its proton for nucleophilic attack. Interestingly, the residues in the binding pocket coordinating $Ca^{2+}$, $Mg^{2+}$, or serine do not show conformational differences in the serine-bound and transition state structure. The only difference is that CDP-DAG is slightly bent towards serine at the β-phosphate and serine in turn is slightly shifted and rotated towards β-phosphate of CDP-DAG. Whether the hydrolysis of the CDP-DAG/serine complex occurs simultaneously with the release of the resulting PSS and CMP (Fig. 6e) cannot be determined from the structures here. As we could not crystallize MjPSS without CDP-DAG (Fig. 6a), we assume that the apo state is flexible or unstable after release of the products and MjPSS quickly binds the next CDP-DAG (Fig. 6b). Another possibility would be that the hydrolysis of the transition state and the release of the products is induced by the next CDP-DAG molecule. Once the next CDP-DAG is bound, the reaction cycle can start again. As MjPSS did not crystallize in the absence of CDP-DAG but of serine (Fig. 6b), we assume that first CDP-DAG and the metal ions bind to the protein and then the serine binds to the pre-formed binding pocket (Fig. 6c). As MjPSS is also able to carry out the reaction reversibly, it should also be able to bind to PS and CMP, and reach the transition state.

In our proposed reaction mechanism, the chloride anions bound near the substrates play a special role in activating the substrate serine for the nucleophilic attack. Bound anions are also present at the corresponding positions of the other structurally known CDP-APs and the residues coordinating them are highly conserved in PSS proteins as well as in the other, general CDP-APs (Supplementary Figs. 12 and 13). In MtPgsA1, it was speculated that the bound citrate occupies the position of D-myo-inositol-3-phosphate[32], which is the molecule analogous to serin in MjPSS. As in our MjPSS structures all substrates are present in the binding pockets and the residues coordinating the anions are highly conserved in the CDP-APs, we assume that these anions play a crucial role in the corresponding reaction catalyzed by these enzymes. Moreover, in AfDIPPS, Af2299, RsPIPS, and MtPgsA1, the anions are coordinated by conserved residues of both monomers within the dimer, suggesting that the activity of these enzymes depends on dimerization or that dimerization depends on binding of anions and/or of substrates.

The structural and functional results of the present work will further contribute to the understanding of the molecular processes of lipid synthesis in various cells, especially those of human pathogenic organisms of medical relevance.

## Methods

**Cloning, protein expression and purification.** The gene encoding PS synthase from *M. janaschii* (MjPSS) was cloned into a plasmid vector (pSKB2LNB) containing an N-terminal $His_6$-Tag and a protease recognition site for PreScission cleavage. *E. coli* C41-(DE3) cells were transformed with the plasmid and grown in auto-induction medium containing 100 mg/L of kanamycin at 309 K for 18 h. Cells were collected by centrifugation at 10,000 r.c.f. for 10 min at 276 K, resuspended in 50 mM Tris-HCl pH 8.0, 100 mM NaCl, and broken with a cell disruptor (Constant System Ltd) at a pressure of 1.9 kBar. Unbroken cells and cell debris were removed by centrifugation at 15,000 r.c.f. for 30 min. Membranes were isolated by centrifugation at 150,000 r.c.f. for 90 min and resuspended in 50 mM Tris-HCl pH 8.0, 50 mM NaCl, 30% glycerol. The protein was solubilized by adding 1% *n*-decyl-β-D-maltopyranoside (DM), 1 M NaCl, 10 mM $MgCl_2$, and 10 mM imidazole to the membrane suspension. Insoluble material was removed by ultracentrifugation at $100,000 \times g$ for 60 min.

The supernatant was applied to a chelating Sepharose column (GE) loaded with $Co^{2+}$ and washed with 2 cv 50 mM Tris-HCl pH 8.0, 1.5 M NaCl, 10 mM $MgCl_2$, 10 mM imidazole, and 0.2% DM followed by 2 cv 50 mM Tris-HCl pH 8.0, 50 mM

NaCl, 10 mM $MgCl_2$, 10 mM imidazole, and 0.2% DM. The protein was eluted with 300 mM imidazole and concentrated to 50 mg/ml using Amicon Ultracel concentrator (30 kDa cut-off). The concentrated protein was applied to a Superose 6 10/300 GL size-exclusion column equilibrated with 20 mM Tris/HCl pH 8.2, 150 mM NaCl. The MjPSS-containing fractions were pooled, concentrated as described above, frozen in liquid nitrogen, and stored at 252 K.

The SeMet substituted protein was expressed in M9 minimal medium. For this, the cells were grown until an $OD_{600}$ of 0.8 and supplemented with a mixture containing 100 mg/l lysine, phenylalanine, threonine, isoleucine, leucine, valine, and SeMet to inhibit the methionine synthesis. After 15 min, the protein was expressed overnight at 297 K after addition of isopropyl β-D-1-thiogalactopyranoside (IPTG) (1 mM final concentration). Purification was performed as described above for native MjPSS, except that a Superdex 200 Increase 10/300 GL column was used in the final size-exclusion chromatography.

**Site-directed mutagenesis.** Site-directed mutagenesis was performed with the QuikChange site-directed mutagenesis kit (Stratagene) according to the manufacturer's instructions using the wild-type MjPSS construct as template and the oligonucleotides listed in the supplementary table 1. All constructs were verified by nucleotide sequencing. The $MjPSS_{\Delta 1-8}$ was amplified and cloned with NdeI and XhoI into the pSKB2LNB vector. The MjPSS mutants were expressed in TB medium. Cells were grown at 309 K until they reached an $OD_{600}$ of 1.4, and the protein was expressed overnight at 297 K after induction with 1 mM IPTG. Purification was as described for native MjPSS, except that a Superdex 200 Increase 10/300 GL column was used for the final size-exclusion chromatography.

**Monoolein preparation.** CDP-DAG (Avanti polar lipids) in chloroform was added to solid monoolein (NuCheck) at a ratio of 2% (w/w) in glass vials until monoolein was completely dissolved. The solution was mixed and incubated at room temperature. After 2–16 h, chloroform was removed by flushing the vial overnight with dry nitrogen.

**Activity measurement of MjPSS in monoolein.** MjPSS at a concentration of 40 mg/ml, with or without the addition of 0.5 mM serine, was mixed with CDP-DAG-doped monoolein in a Hamilton syringe. The mixture was transferred to a glass vial and incubated at 309 K overnight. The protein/monoolein mixture was resuspended in 100 µl of water, supplemented with 375 µl of methanol:chloroform (2:1 v/v) solution, and mixed for 20 s. After addition of another 100 µl chloroform and 100 µl water, the mixture was mixed again and centrifuged at 20,000 r.c.f. for 3 min. The lipid-containing organic phase at the bottom of the tube was transferred to a clean tube and the chloroform was evaporated under dry nitrogen. The dry lipids where resuspended in 10 µl of fresh chloroform and applied on a HPTLC silica plate. The mobile phase was composed of chloroform:methanol:water in a ratio of 65:25:4. For the measurement of the reverse reaction, 25 µl of POPS (20 mM) was mixed with 4 µl of MjPSS (5 mg/ml), 12 µl of CMP (10 mM), and 9 µl of buffer (50 mM Tris pH 8, 50 mM NaCl, 0.2% (w/V) DM). The mixture was incubated at 342 K for 180 min and diluted with 100 µl water. The lipids were extracted and analyzed by HPTLC.

**HPTLC analysis.** Next, 25 µl reaction mixture was diluted to a volume of 100 µl by addition of water and mixed for 20 s with 375 µl of a mixture of methanol and chloroform (2:1 v/v). After addition of 100 µl chloroform and 100 µl water, the mixture was mixed again and centrifuged at 20,000 r.c.f. for 3 min. The lipid-containing organic phase at the bottom of the tube was transferred to a clean tube and the chloroform was evaporated under dry nitrogen. The dry lipids where resuspended in 10 µl of chloroform and applied on a HPTLC silica plate. The mobile phase was composed of chloroform:methanol:water in a ratio of 65:25:4.

**Activity measurement of MjPSS by HPLC.** The reaction mixture of the activity assay contained 2 µl of MjPSS at 5 mg/ml, 12.5 µl of 2 mM CDP-DAG, varying amounts of 15 mM serine as described for each experiment, and buffer consisting of 50 mM Tris-HCl pH 7.0, 50 mM NaCl, 0.2% DM. The sample was incubated for different time intervals at different temperatures and the reaction was stopped by denaturing the protein by heating the mixture in the thermocycler at 371 K for 10 min. The denatured protein was removed by centrifugation and the supernatant was diluted by adding 20 µl of the reaction mixture to 90 µl of HPLC running buffer. Of this solution, 100 µl was injected into the HPLC.

The HPLC running buffer consisted of 100 mM potassium phosphate pH 6.6, 10 mM EDTA, 10 mM tetra butyl-ammonium bromide and 5% methanol. The column used was a Supelco Zorbax ODS 25 cm, 4.6 mm with a flow rate of 0.9 ml/min.

**Binding analysis by microscale thermophoresis.** Binding analysis of substrates to MjPSS by microscale thermophoresis was performed using a Monolith device (NanoTemper Technologies GmbH). The final protein concentration was 50 nM in 50 mM HEPES pH 8, 50 mM NaCl, 0.2% (w/V) DM. Divalent cations that remained bound to MjPSS were removed by incubating with EDTA for 6 h and dialyzing overnight against EDTA-free buffer (50 mM HEPES pH 8, 50 mM NaCl,

0.2% (w/V) DM). The dialyzed protein solution was diluted to a concentration of 100 nM. For protein labelling, the Monolith His-Tag labelling Kit RED-tris-NTA was used at a final concentration of 25 nM. The ligands tested were all dissolved in the same buffer as the protein at a stock concentration of 2 μM. All experiments were performed with a working concentration of 1 μM. Experiments were performed after 1 h of incubation at room temperature. Raw data were exported and normalized fluorescence for all samples was compared to the MjPSS standard after 5 s of scanning. Variance analysis of the normalized fluorescence with a confidence interval of 2% gave an indication of binding.

**Crystallization of MjPSS in lipidic cubic phase**. LCP was used for initial crystallization screens. Monoolein alone or with the addition of 2% (w/w) of CDP-DAG (Avanti polar lipids) was used for this purpose. For the screening experiments, MjPSS (40 mg/ml) and the monoolein were mixed manually in a 3 : 1 ratio using a Hamilton syringe at room temperature until a clear and homogeneous phase was obtained. Commercially available crystallization solutions (e.g., Qiagens MBclass Suit and MBclass Suit II Molecular Dimension MemGold) were used for the initial crystal screenings. A Mosquito robot (TTPlabtech) was used to dispense 50–100 nl LCP phase and 500 nl of precipitating solution. The glass sandwich plates were incubated at 292 K.

**Crystallization of MjPSS in lipidic sponge phase**. The monoolein lipid mixture was mixed with the reservoir solution in ratio of 3 : 1 without adding a precipitant. The resulting LCP was transferred to a glass vial and a fourfold excess of the reservoir solution with added precipitant was layered on top of the LCP. The vial was flushed with N2, sealed and incubated for several days until two clear phases separated. Lipidic sponge phase (LSP) screenings were performed by pipetting 1 μl of the protein solution onto a cover slide and overlaying it with 4 μl of the clear LSP solution. Screening was performed in 24 hanging drop vapour diffusion plates. Next, 500 μl of the sponge phase inducing solution was used as a reservoir with the addition of various salt concentrations ranging from 400 mM to 1 M NaCl. The crystals were were snap-frozen in liquid nitrogen.

**Data acquisition and processing**. The diffraction data were collected at the Swiss Light Source Beamline PXII (Paul Scherrer Institute, Villigen, Switzerland) and at the Petra-3 beamline P11 (DESY, Deutsche Elektron-Synchrotron) under a constant stream of cold nitrogen gas (100 K). Data processing, integration, and scaling was performed with the XDS package[45]. For phase determination, protein labelled with SeMet was expressed, purified, and screened for crystals. All crystals grown by in meso crystallization and belonged to the orthorhombic space group $P2_12_12$ or C2 (Table 1). High-quality crystals were obtained with native protein in the absence and presence of $Ca^{2+}$ and with SeMet-labelled protein with CDP-DAG and serine. The single-wavelength anomalous data collected from SeMet crystals were used to determine the phases and to solve the structure. The initial experimental electron density was refined by density modification using resolve[46]. The protein backbone was manually traced in COOT[47] and used as starting model for automatic chain tracing in Buccaneer[48]. The resulting model was used as search model in phenix.phaser-MR[49,50] for molecular replacement using the native data. All models were iteratively rebuilt into $2F_o − F_c$ and $F_o − F_c$ electron density maps using COOT[47] and refined using the phenix.refine subroutine from the PHENIX programme suite[51]. The data collection, refinement, and model statistics are summarized in Table 1. The figures were generated with ChimeraX[52]. pH-dependent pKa values were calculated using pdb2pqr[53].

**Reporting summary**. Further information on research design is available in the Nature Research Reporting Summary linked to this article.

## Data availability
The structural data and coordinate files generated in this study have been deposited in Protein Data Bank (PDB) with the accession codes 7B1K for closed state with CDP-DAG and one citrate, 7B1L for the open state with CDP-DAG and two serines, 7POW for the open state with CDP-DAG/serine complex in the transition state, and 7B1N for the closed state with CDP-DAG. The data that support the findings of this study are available from the corresponding author upon request. Source data are provided with this paper.

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

## Acknowledgements

We thank Werner Kühlbrandt for continuous support, encouragement, and critical comments on the manuscript, David Wöhlert for helpful discussions, and Sabine Häder and Christina Kunz for excellent laboratory management. Furthermore, we thank the staff at the PXII beamline at the Swiss Light Source, SLS (Paul Scherrer Institute, Villigen, Switzerland), and the staff of the P11 beamline at Petra-3 of DESY (Deutsche Elektron-Synchrotron in Hamburg) for their support in crystal screening and data collection. We also thank the staff at the beamlines of the European Synchrotron Radiation Facility (ESRF, Grenoble) for their support in crystal screening. This work was funded by the Max Planck Society.

## Author contributions

M.C., H.B., and K.v.P. purified MjPSS. M.C. and K.v.P. screened for crystals. M.C. performed functional analysis studies and crystallized MjPSS in LCP and LSP. M.C. and O.Y. collected diffraction data, solved, and analyzed the structures. M.C. and O.Y. wrote the manuscript. O.Y. coordinated and supervised the project.

## Funding

## Competing interests

The authors declare no competing interests.
