## [Peer Review File · Nature Communications]

Crystal structures of phosphatidyl serine synthase PSS reveal the catalytic mechanism of CDP-DAG alcohol O phosphatidyl transferasesREVIEWER COMMENTS

Reviewer #1 (Remarks to the Author):

Martin et al. have solved for structures of the integral membrane enzyme CDP-DAG alcohol O-phosphatidyl 2 transferases from archaea (MjPSS), in Close states and Open states. Structures of integral membrane enzymes remain largely unresolved in the PDB, even amongst membrane proteins which themselves are underrepresented compared to soluble proteins, making any new structures of intrinsic interest. In addition to the amounts of structural work, the authors have characterized the function of the enzyme and defined the substrate specificity by biochemical experiment. Overall, this is a well-performed structural study on understanding the class of transmembrane proteins. It should be addressed all of my questions before acceptance. Some specific comments and queries here also are raised, outlined below.

1. The author should perform binding experiments between the enzyme and substrates, furthering to verify its specificity.
2. In Fig 1 and other similar supplementary figures should combine with spot plot.
3. Fig 1 and 4 - please consider changing the colors – Fig 1 looks plain, which could be highlighted by colored, in addition, MjPSS catalytic scheme could be presented; and Fig. 4 is not very colorblind friendly.
4. In table 1, PDB 7B1L, the completeness in higher resolution is only 88.9%, the resolution may be over claimed;
5. In the refine data of PDB 7B1N, the gap between Rwork and Rfree is over 5, the final structure could try refine better. Moreover, values of Ramach. favored (%) of all PDB seem a bit higher, more accurate refinement needs.
6. Line 180-193 Details of crystallization and structures determination move into the methods.
7. Line 193-195 this presentation is wrong. First, the molecular weight of Membrane proteins could be induced from gel-filtration, because of integrated proteins in complex with detergents to form micelle; second, the asymmetric unit has two MjPSS molecules have to be verified it was not the crystal packing, but it has activity in physiology or in vitro. Please make this more explicit if it is still as the author state.
8. The CDP-DAG-bound MjPSS structure, which similar to other lipids synthase, but not limited in the text introduction and discussion, it should illustrate more extensively in introduction and discussion.
9. Are there any other conformational changes observed among these structural states besides Helix 7? If not, please discussion why.
10. Please show for all structures a figure demonstrating representative density quality clearly.
11. Given how the field is, the introduction would benefit from more text focused on lipid or lipid-like biosynthesis, to attract more audiences.

Reviewer #2 (Remarks to the Author):

Centola et al. describe the structural and functional characterization of CDP-AP phosphatidyl serine synthase from *Methanocaldococcus jannaschii* (MjPSS). Four crystal structures were obtained with: (I) CDP-DAG, Ca²⁺, Mg²⁺, and citrate, (II) CDP-DAG, two serines, Ca²⁺, Mg²⁺, (III) CDP-DAG/serine complex, Ca²⁺, and Mg²⁺, and (IV) CDP-DAG and Ca²⁺.

To date, MjPSS is the only structure solved with both substrates. It is indeed a pity not to have been able to characterize the PSS structure without CDP-DAG to elucidate the “apo” form and assess if structural rearrangements occur upon ligand binding.

Based on these structural data along with enzymatic assays, a catalytic mechanism of CDP-DAG alcohol phosphatidyl transferases is proposed. I was wondering if other experimental assays could be done to corroborate the proposed catalytic mechanism for PSS? Another point that can be improved is to make alignments using a high number of amino acid sequences or even against databases to obtain significant results.

Four previous publications on the 3D structure of members from the same family of enzymes (CDP-AP transferases) have already been reported. The di-myo-inositolphosphate phosphatase synthases or

similar (AfDIPPS, Af2299) (DIPPSs) and phosphatidylinositolphosphate synthases (PIPSs).

Comments:

The authors propose a bi-bi reaction mechanism of a second order nucleophilic attack since both substrates, CDP-DAG and serine, are required for the proper activity of MjPSS. Because MjPSS did not crystallise in the absence of CDP-DAG but of serine, they suggest that CDP-DAG and the metal ions bind first to the protein and that afterwards the serine binds to the preformed binding pocket. Assuming a sequential mechanism, it'd be interesting to investigate whether synthesis of PS is random or ordered, e.g. if the order of substrate binding is relevant. The fact that crystals with serine were only obtained when CDP-DAG was present may be a hint, but cannot exclude a random reaction. Since the authors already have a titration with serine on MjPSS with CDP-DAG, assays with addition of CDP-DAG on MjPSS previously incubated with serine could provide some insights into this.

Based on the high sequence conservation of the residues surrounding the Cl⁻ anions and their proximity to the catalytically active serine and Mg²⁺ (Suppl. Fig. 5) the authors suggest an important role for these ions, corroborated by decrease/loss of activity on mutants (interacting arginine residues). I strongly recommend sequence alignments with higher number of PSS from different sources (besides the 5 selected organisms) to evaluate the conservation of residues, especially to assess the ones involved in substrates and Cl binding.

The discussion on the comparison of MjPSS with other CDP-APs (~DIPPS or PIPS), whose structures have been reported, namely the number of TMHs and ligand binding would be enriched from exploring the output from broader seq. alignments with other PSSs. Is the predicted number of TMHs in other PSSs also 8? Any additional rationale for 8 TMHs instead of 6 for PIPs/DIPSSs? Can these predicted number be hinted from the sequence alignments? Is there any conservation at the dimeric interface and ligand binding? Do the substrates influence the different interface of PSS, PIP or DIP? Any plausible explanation for the different position of the acyl chains of CDP-DAG in MjPSS and RsPIP?

An additional Supl. Figure zooming the active sites with bound ligands/ions from the various 3D structures may help to better understand their structural comparison.

I'd also have liked to see a thorough discussion on how the herein proposed catalytic mechanism for PSSs compare with the previous ones for PIPS, bearing in mind the common CDP-DAG and different alcohol substrates.

Other points to review:

The difference between AfDIPPS and Af2299 is not correct and sometimes misused. AfDIPPS relates to *A. fulgidus* DIPPS (enzyme AF0263, Nogly et al.) that catalyzes the condensation of CDP-inositol with inositol-1-phosphate to produce di-inositol-1,3'-phosphate-1'-phosphate (DIPP), while the activity of AF2299 was not reported, it's proposed to use CDP-glycerol instead yielding glyceryl phosphodiester of inositol-1-phosphate (Sciara et al.).

Check spelling, you sometimes use English (UK or USA styles), as for ex crystallisation/crystallization, synthesise/synthesize, use of "coma" instead of "point", ex. Vmax of 2,22 μM/s, use °C instead of K (international unit of T), the word "why" is sometimes misplaced.

Page 7- Not all groups belonged to the orthorhombic space group P21212, one dataset is monoclinic C2.

Page 8- You mention size-exclusion chromatography and SDS-PAGE analysis to indicate MjPSS as a dimer, but SEC is quite tricky for membrane proteins solubilized in detergents due to their micellar size, in this respect SEC-MALS is more appropriate. I understand the crystallographic interface corroborates the dimeric arrangement.

Supl. Fig. 3- This figure gives a nice overview on the various MjPSS structures I'd suggest try use of transparency for secondary structure elements or different colors for Mg and Ca, and thicker/darker sticks for CDP-DAG to better highlight ligands and ions, which are difficult to visualize.

The statistics in Table 1 looks fine, I was just wondering if there is any explanation why RMSD for bonds and angles is much higher for the C2 structure compared to the others.

Reviewer #1 (Remarks to the Author):

Martin et al. have solved for structures of the integral membrane enzyme CDP-DAG alcohol O-phosphatidyl 2 transferases from archaea (MjPSS), in Close states and Open states. Structures of integral membrane enzymes remain largely unresolved in the PDB, even amongst membrane proteins which themselves are underrepresented compared to soluble proteins, making any new structures of intrinsic interest. In addition to the amounts of structural work, the authors have characterized the function of the enzyme and defined the substrate specificity by biochemical experiment. Overall, this is a well-performed structural study on understanding the class of transmembrane proteins. It should be addressed all of my questions before acceptance. Some specific comments and queries here also are raised, outlined below.

We thank reviewer #1 for the in overall positive feedback and comments. We addressed all questions of the reviewer in the revised manuscript. We are confident that the revised manuscript is considerably improved now.

#1.1

The author should perform binding experiments between the enzyme and substrates, furthering to verify its specificity.

We performed binding analysis of MjPSS to its substrates and potential substrates by micro-scale thermophoresis (MST) and show that the change in fluorescence by CDP-DAG and serine binding is significantly higher than the fluorescence change for the other potential substrates. The result and the method are integrated into the revised manuscript text and the corresponding graph is shown in the supplementary figure 2B.

#1.2

In Fig 1 and other similar supplementary figures should combine with spot plot.

In the revised manuscript we used spot plots in the corresponding figures.

#1.3

Fig 1 and 4 - please consider changing the colors – Fig 1 looks plain, which could be highlighted by colored, in addition, MjPSS catalytic scheme could be presented; and Fig. 4 is not very colorblind friendly.

In the revised manuscript we made figure 1 completely black & white. For overall consistency, we would prefer to keep the colors in figure 4.

#1.4

In table 1, PDB 7B1L, the completeness in higher resolution is only 88.9%, the resolution may be over claimed;

The completeness for the high-resolution shell of 7B1L is 95.6%. We assume the reviewer means 7B1M. While the completeness for the highest-resolution shell of 7B1M is slightly lower, the other statistical values are still acceptable, why we decided to set the resolution cut-off to that value.

#1.5

In the refine data of PDB 7B1N, the gap between Rwork and Rfree is over 5, the final structure could try refine better. Moreover, values of Ramach. favored (%) of all PDB seem a bit higher, more accurate refinement needs.

The gap between Rwork and Rfree was below 5 in most of the previous refinement cycles and jumped slightly above 5 in the final step. However, as there are no obvious differences between the previous refined structures and the final structure, we do not see the need for further refinements.

#1.6

Line 180-193 Details of crystallization and structures determination move into the methods.

Done

#1.7

Line 193-195 this presentation is wrong. First, the molecular weight of Membrane proteins could be induced from gel-filtration, because of integrated proteins in complex with detergents to form micelle;

In general, the reviewer is right. We compared the elution profiles with other membrane proteins of similar size and detergent environment. Nevertheless, in the revised manuscript we have removed the relevant text passage and included the SDS-PAGE analysis in the supplementary figure 3C, which clearly shows MjPSS as dimer, even in the presence of SDS. MjPSS only runs as a monomer only when the sample is heated to 95°C for 5 minutes in the presence of SDS. We also rule out the interaction of monomers by disulfide formation, as the presence of β -mercaptoethanol or DTT shows no difference in the SDS-PAGE analysis. This information is also included in the revised manuscript and shown in supplementary figure 3.

second, the asymmetric unit has two MjPSS molecules have to be verified it was not the crystal packing, but it has activity in physiology or in vitro. Please make this more explicit if it is still as the author state.

We show detergent-solubilised MjPSS as a dimer in SDS-PAGE and assume that this is the *in vivo* state of MjPSS in the membrane.

#1.8

The CDP-DAG-bound MjPPS structure, which similar to other lipids synthase, but not limited in the text introduction and discussion, it should illustrate more extensively in introduction and discussion.

Done

#1.9

Are there any other conformational changes observed among these structural states besides Helix 7? If not, please discussion why.

The most significant difference between the open and closed states is the position of the N-terminal part of helix 7. The other minor differences, e. g. the C-terminal part of helix 2 and 8, and the differences in the loops are shown in the supplementary figure 6, which was apparently overlooked by the reviewer (Suppl. Fig. 5 in the original manuscript).

#1.10

Please show for all structures a figure demonstrating representative density quality clearly. In addition to figure 5, which shows the representative electron density for CDP-DAG, we produced another supplementary figure in the revised manuscript showing the interface between one monomer and the N-terminal helix from the other monomer.

#1.11

Given how the field is, the introduction would benefit from more text focused on lipid or lipid-like biosynthesis, to attract more audiences.

Done

Reviewer #2 (Remarks to the Author):

Centola et al. describe the structural and functional characterization of CDP-AP phosphatidyl serine synthase from *Methanocaldococcus jannaschii* (MjPSS). Four crystal structures were obtained with: (I) CDP-DAG, Ca²⁺, Mg²⁺, and citrate, (II) CDP-DAG, two serines, Ca²⁺, Mg²⁺, (III) CDP-DAG/serine complex, Ca²⁺, and Mg²⁺, and (IV) CDP-DAG and Ca²⁺.

To date, MjPSS is the only structure solved with both substrates. It is indeed a pity not to have been able to characterize the PSS structure without CDP-DAG to elucidate the “apo” form and assess if structural rearrangements occur upon ligand binding.

We fully agree with the reviewer. The apo-structure of MjPSS would contribute to a better understanding of the mechanisms of this enzyme family. Unfortunately, although we were able to crystallise MjPSS in the absence of the substrates, the crystals did not diffract better than 10Å, which is not sufficient for structure determination.

#2.1

Based on these structural data along with enzymatic assays, a catalytic mechanism of CDP-DAG alcohol phosphatidyl transferases is proposed. I was wondering if other experimental assays could be done to corroborate the proposed catalytic mechanism for PSS?

micro-scale thermophoresis (MST), we determined the binding of substrates and substrate-like molecules to MjPSS. Furthermore, we were able to show experimentally that MjPSS can perform the reverse reaction when the products PS and CMP are present in high concentrations. Further support for the proposed mechanism comes from the fact that MjPSS cannot catalyse the formation of CDP-DAG from PA and CMP. The results have been incorporated into the revised manuscript. We reserve further and more detailed experiments for further studies on this interesting enzyme.

#2.2

Another point that can be improved is to make alignments using a high number of amino acid sequences or even against databases to obtain significant results.

In the revised manuscript, we compare 13 PSS sequences from bacterial, archaeal and eukaryotic organisms.

#2.3

Four previous publications on the 3D structure of members from the same family of enzymes (CDP-AP transferases) have already been reported. The di-myoinositolphosphate phosphate synthases or similar (AfDIPPS, Af2299) (DIPPSs) and phosphatidylinositolphosphate synthases (PIPSs).

Yes, the reviewer is right. In the manuscript we refer to these structures in the introduction, and compare them with our structures (see supplementary figure 11 and 12)

Comments:

#2.4

The authors propose a bi-bi reaction mechanism of a second order nucleophilic attack since both substrates, CDP-DAG and serine, are required for the proper activity of MjPSS. Because MjPSS did not crystallise in the absence of CDP-DAG but of serine, they suggest that CDP-DAG and the metal ions bind first to the protein and that afterwards the serine binds to the preformed binding pocket. Assuming a sequential mechanism, it'd be interesting to investigate whether synthesis of PS is random or ordered, e.g. if the order of substrate binding is relevant. The fact that crystals with serine were only obtained when CDP-DAG was present may be a hint, but cannot exclude a random reaction. Since the

authors already have a titration with serine on MjPSS with CDP-DAG, assays with addition of CDP-DAG on MjPSS previously incubated with serine could provide some insights into this.

We have performed CDP-DAG titration against MjPSS pre-incubated with serine, but cannot determine whether the reaction is sequential or random because the molecules in our detergent lipid mixture are free to rearrange, so we have no control on the binding sequence. Using micro-scale thermophoresis, we show that MjPSS is able to bind CDP-DAG and serine independently. We also show that MjPSS is able to catalyse the reverse reaction in the presence of its products PS and CMP at higher concentrations. The fact that CDP-DAG is not formed from PA and CMP further supports the proposed mechanism. We were unsuccessful in titrating of CDP-DAG to serine-preincubated MjPSS. We believe that the results of the experiments are of little or no significance due to diffusion effects. As stated before, the results of the additional experiments are included in the revised manuscript.

#2.5

Based on the high sequence conservation of the residues surrounding the Cl⁻ anions and their proximity to the catalytically active serine and Mg²⁺ (Suppl. Fig. 5) the authors suggest an important role for these ions, corroborated by decrease/loss of activity on mutants (interacting arginine residues). I strongly recommend sequence alignments with higher number of PSS from different sources (besides the 5 selected organisms) to evaluate the conservation of residues, especially to assess the ones involved in substrates and Cl binding.

In the revised manuscript, we compare 13 PSS sequences from bacterial, archaeal and eukaryotic organisms. The conservation of the chloride-coordinating residues is also clearly visible in the new alignment with the additional PSS sequences. In the revised manuscript, we compare the sequences of the structurally known CDP-APs in the new sequence alignment in supplementary figure 13. Here, we highlighted the CDP-AP sequence motif and the anion coordinating residues analogous to Arg101 and Arg104. We excluded MjPSS from this alignment because the sequence similarity is too low. Interestingly, in the compared dimeric structures, the anions are coordinated not only from one protomer, also residues from the second protomer contribute in their coordination. These residues are highlighted by purple arrows in the new sequence alignment (supplementary figure 13).

#2.6

The discussion on the comparison of MjPSS with other CDP-APs (~DIPPS or PIPS), whose structures have been reported, namely the number of TMHs and ligand binding would be enriched from exploring the output from broader seq. alignments with other PSSs.

The prediction of transmembrane helices for PSS proteins proved not to be very reliable, as the following prediction for MjPSS shows.

The first and last two helices in MjPSS are predicted to be a single TMH. Helix 3, which (together with the C-terminal part of helix 2) coordinates the substrates, is also not

correctly predicted as a TMH. On the other hand, the sequence alignment shows greater variability not only for the C-terminal helices, but also the N-terminal regions vary between different PSS proteins. Whether the N-terminal regions form transmembrane helices and fulfill the functionality of helix 7 and 8 of MjPSS needs to be shown structurally.

#2.7

Is the predicted number of TMHs in other PSSs also 8? Any additional rationale for 8 TMHs instead of 6 for PIPs/DIPs?

Most secondary structure prediction tools/server predict 4-6 TMHs for the PSS proteins (see #2.6) that we used for the new sequence alignment. Further analysis and comparison of the PSS/DIPS/PIP/... structures would be required to make reliable statements about the additional helices of MjPSS, but this is beyond the scope of the present work.

#2.8

Can these predicted number be hinted from the sequence alignments? Is there any conservation at the dimeric interface and ligand binding?

No, we cannot say from the sequence alignment whether the C-terminal or N-terminal regions of the PSS proteins are TMHs because these regions are not conserved. The residues that form the dimer interface, in the case of MjPSS, are on the opposite side of the helix with the substrate-coordinating residues. Helix 1 and 3 (Thr9 to Leu28 and Phe57 to Tyr79) and the back side of the chloride-coordinating residues in helix 4 (Phe94 to Asn106) show higher conservation in the PSS sequences.

#2.9

Any plausible explanation for the different position of the acyl chains of CDP-DAG in MjPSS and RsPIP?

In MjPSS, the acyl chains have to fit the hydrophobic cavity within the molecule, whereas in RsPIP the acyl chains are exposed to the membrane lipids, probably similar to the situation *in vivo*. There, the acyl chains can theoretically adopt any conformation within the hydrophobic region. However, we do not want to speculate whether the situation is different in the crystallization environment of RsPIP in the lipidic cubic phase, restricting the acyl chains to the conformations observed in the structure

#2.10

An additional Supl. Figure zooming the active sites with bound ligands/ions from the various 3D structures may help to better understand their structural comparison.

We have reorganized the supplementary figures and show an overall overlay of the structures in supplementary figure 11. We also show the substrate binding site with the substrates for MjPSS and the corresponding CDP-AP in higher magnification in the new supplementary figure 12.

#2.11

I'd also have liked to see a thorough discussion on how the herein proposed catalytic mechanism for PSSs compare with the previous ones for PIPS, bearing in mind the common CDP-DAG and different alcohol substrates.

In the revised manuscript, we compare the reaction mechanism to the other CDP-APs and discuss them in more detail.

Other points to review:

#2.12

The difference between AfDIPPS and Af2299 is not correct and sometimes misused. AfDIPPS relates to A. fulgidus DIPPS (enzyme AF0263, Nogly et al.) that catalyzes the condensation of CDP-inositol with inositol-1-phosphate to produce di-inositol-1,3'-phosphate-1'-phosphate (DIPP), while the activity of AF2299 was not reported, it's proposed to use CDP-glycerol instead yielding glyceryl phosphodiester of inositol-1-phosphate (Sciara et al.).

We thank the reviewer for the clarification. In the structural alignments we used the CDP-AP, which contain a ligand (Af2299, pdb 4q7c; MtPgsA1, pdb 6h59 and RsPIPS, pdb 5d92) and excluded AfDIPS. Therefore, we corrected the labelling in the text and figures accordingly.

#2.13

Check spelling, you sometimes use English (UK or USA styles), as for ex crystallisation/crystallization, synthesise/synthesize, use of "coma" instead of "point", ex. Vmax of 2,22 μ M/s, use $^{\circ}$ C instead of K (international unit of T), the word "why" is sometimes misplaced.

Spelling checked and corrected.

#2.14

Page 7- Not all groups belonged to the orthorhombic space group P21212, one dataset is monoclinic C2.

Corrected.

#2.15

Page 8- You mention size-exclusion chromatography and SDS-PAGE analysis to indicate MjPSS as a dimer, but SEC is quite tricky for membrane proteins solubilized in detergents due to their micellar size, in this respect SEC-MALS is more appropriate. I understand the crystallographic interface corroborates the dimeric arrangement.

In general, the reviewer is right. We compared the elution profiles to other membrane proteins of similar size and detergent environment. Nevertheless, in the revised manuscript we have removed the corresponding text passage and included the SDS-PAGE analysis in the supplementary figure 3C, which clearly shows MjPSS as dimer. even in the presence of SDS. MjPSS runs as monomer only if the sample is heated to 95 $^{\circ}$ C for 5 minutes in presence of SDS. We also exclude the interaction of monomers by disulfide formation, as the presence of β -mercaptoethanol or DTT shows no difference in the SDS-PAGE analysis.

#2.16

Supundpl. Fig. 3- This figure gives a nice overview on the various MjPSS structures I'd suggest try use of transparency for secondary structure elements or different colors for Mg and Ca, and thicker/darker sticks for CDP-DAG to better highlight ligands and ions, which are difficult to visualize.

We used tubes for the helices and highlighted the ligands by making the sticks thicker in the supplementary figure 4 of the revised manuscript.

#2.17

The statistics in Table 1 looks fine, I was just wondering if there is any explanation why RMSD for bonds and angles is much higher for the C2 structure compared to the others. The referee is right. The RMSDs for bonds and angles are slightly higher than for the other structures. This dataset was collected with a very low flux and processed anomalously. Nevertheless, we performed a more careful refinement of the final structure using this dataset and slightly improved the statistical values. The overall structure of the protein and

the ligands did not change significantly in this process. We uploaded the re-refined structure in the PDB. The resulting validation reports are also uploaded to the manuscript upload system of the journal. The new PDB entry for this structure is 7POW and replaces the (unreleased) entry 7B1M.

REVIEWER COMMENTS

Reviewer #1 (Remarks to the Author):

Happy to review the revised manuscript, no more concerns to be raised. but minor issue, such as: there are no recently published paper as references for the advance of lipid biosynthesis. Once this is addressed I would strongly recommend this manuscript for publication in Nature communications.

Reviewer #2 (Remarks to the Author):

The authors made an effort to meet the request from the reviewers. I'm satisfied with the authors' responses and revised manuscript, which has been improved.

Reviewer #1 (Remarks to the Author):

Happy to review the revised manuscript, no more concerns to be raised. but minor issue, such as: there are no recently published paper as references for the advance of lipid biosynthesis. Once this is addressed I would strongly recommend this manuscript for publication in Nature communications.

We thank reviewer #1 for the in overall positive feedback and comments. We addressed the remaining points of the reviewer in the revised manuscript and mentioned the recent structure of MkPIPS in the Introduction and discussed the cardiolipin synthase from the archaeon *Methanospirillum hungatei* in the discussion.

Reviewer #2 (Remarks to the Author):

The authors made an effort to meet the request from the reviewers. I'm satisfied with the authors' responses and revised manuscript , which has been improved.

We thank reviewer #1 for helping us to improve the manuscript